# [Re] Synbols : Probing Learning Algorithms with Synthetic Datasets

## Reproducibility Summary

**Scope of Reproducibility**

This report focuses on the reproduction of some results presented of the above-mentioned paper [3]. Authors introduced a new data generator called Synbols allowing fast generation of low-resolution images rich in latent features. Researchers explored the capabilities of the tool by training popular machine learning algorithms in various M.L paradigms with their synthetically generated datasets. The tool is also trying to address some broader issues relevant to the whole field (i.e. faster iteration cycles for the training, less reliance on expensive hardware, etc.).

To assess the features of Synbols and its capacity to explore well known neural network architectures, we decided to reproduce the results of the Supervised Learning classification task and the Unsupervised Representation Learning experiments. We then generated some datasets with the same attributes to assure the results were consistent. Additionally, we tried to get further insights for the unsupervised task by modifying classifier downstream. The final code used for implementing the replicated results can be found here: [Re] Synbols Repository

**Methodology**

Regarding our methodology, we predominantly followed authors instructions and their publicly available code. Modifications to the original code made in order to further explore some findings will be discussed later in the corresponding section.

**Results**

We manage to reproduce the original results falling within a 2% margin of the reported values. We were pleasantly surprised given the number of models and datasets tested. And thus conclude that Synbols is a well designed tool for rapidly generating a wide variety of low resolution images of UTF-8 characters and strings.

**What was easy / What was difficult**

We applaud authors reproducibility efforts and their availability whenever we had questions. A repository specifically made in order to facilitate the reproduction was available and an up-to-date docker image was also at our disposal to help generate more datasets with the tool. No hidden/forgotten assumptions were needed to reproduce their results. Originally, for the two paradigms tested, twelve different models were trained. Although important hyper-parameters and architecture choices were always mentioned or referenced, we sometimes struggled to check their implementation to see if everything was performed as reported.

**Communication with original authors**

We actively reached out the original authors through e-mail and meeting sessions. The authors always made time to answer our questions. Hereby, we sincerely thank the authors for providing us adequate supports during the reproducibility.

# 1 Introduction

The original paper [3] introduces Synbols, a dataset generator with a rich latent feature space. It generates low resolution images to support quick iteration times. More than 1000 artistic fonts over 14 different languages were collected. The diversity of background and foreground can also vary from solid, gradient, camouflage and natural. Occlusion can also be added to the foreground. In each symbol or character, one can modify the inherent attributes of the image or the character itself. This includes *translation, scale, rotation, shear, bold,* and *italic*. The authors used this versatile tool to probe the limits of existing algorithms in different machine learning paradigms relevant in the field of computer vision. The motivation behind designing a low-resolution dataset generator is that, usually in order to obtain state-of-the-art performance, the model is expected to train on large-scale dataset, especially when the model complexity is high. But it comes at the cost of slow iteration cycles, taking sometimes weeks of training before obtaining the expected results. On the other hand, applying small-scale datasets to train new SOTA models would limit the capability of testing their generalization capacity but also prevent meaningful model comparison. Still, relying on very large datasets creates a high barrier to entry for many organizations and researchers wanting to get into the Deep Learning Revolution [4]. Finally, current research is biased towards fast methods leveraging big datasets instead of considering a more qualitative approach. Synbols aims at solving those issues. Our team is confident that this field of research is of importance for the future and hope that the following reproducibility report will help assess with more confidence the presented claims to allow more research to be conducted on this topic.

Our report is articulated around three key questions ;

- Are the original results reproducible ?
- Were there any hidden assumptions in order to obtain the same results ?
- Can we quickly generate similar datasets ?

# 2 Methodology

In the original article, authors probed six machine learning paradigms in order to test their synthetically generated datasets. Researchers aim was to further investigate strengths and weaknesses of popular machine learning models by exposing them to a wide range of challenging datasets generated by Synbols. We focused our efforts on replicating the supervised learning and the unsupervised representation learning experiments.

In order to facilitate the reproducibility of the experiments and the results presented in the paper, authors made the code used for the benchmarks publicly available. The repository contained the model architectures, the training/testing/validation in HDF5 format storing the images but also the corresponding attributes used in the generation. Each dataset was generated three times using different pseudo-random seed in order to test more thoroughly each dataset. For the more computationally demanding models we ran the experiment using only one seed. We additionally decided to generate the camouflage dataset using the same attributes and seed. The two datasets were identical and provided consistent results. To gain further insights on the unsupervised representation task we edited the source code. More specifically, we modified the classifier downstream on the pipeline by tweaking the original MLP and then trying with a linear regression. We tried implementing a different classifier (EfficientNet) but it did not provide any meaningful insight to understanding the low performance in the unsupervised task.

# 3 Reproducibility resources

The computational resources required to reproduce the experiment were very accessible. Authors originally used Tesla V100 (TDP of 300W) type hardware for a cumulative 23916 hours of computation needed for the whole paper (this includes debugging, failed experiments and hyperparameter search). By focusing on two experiments and reducing the number of seed tested, we were able to reproduce their results in approximately 194 hours using a Tesla K80 (TDP of 300W) type GPU with 12GB of GDDR5 memory available on Google Cloud Platform. Total emissions are estimated to be 1.16 kgCO2eq. [2]. All models were implemented using Pytorch.

## 3.1 Datasets

### 3.1.1 Supervised Learning

The Synbols default dataset will serve as baseline for other dataset and it consists of samples of English characters with a font uniformly selected from the font collection and the attributes are selected to have high variance. Respectively for

the Camouflage and Natural datasets, the according feature was added to the default dataset. The Less Variations dataset removes the italic and bold attributes and reduces the variations of other attributes. Finally, the Korean dataset consists of a uniformly selected Hangul characters (reduced to the first 1000 symbols). The width and height and channels of all of the images is 32x32x3 and the dataset size was 100k[1]. The authors also decided to confront those synthetic datasets to popular benchmark datasets, namely MNIST and SVHN. We did not reproduce the results for those standard datasets instead choosing to focus our efforts on the synthetic datasets generated by the tool.

### 3.1.2 Unsupervised Representation Learning

In the Unsupervised Representation Learning, the paper leverages three variants of datasets, namely, solid, camouflage, and shades. In these datasets the bold attribute was kept on while a low variance was applied on the scale. The first variant, the solid dataset used black and white contrast while a smooth gradient was applied on the shade variant. In the camouflage dataset the corresponding attribute was added. The width and height and channels of all of the images is 32x32x3. Moreover, due to limited resources we only used one of the three variant of each dataset [2].

## 4 Model Architecture

All the models were trained using adaptive learning rate optimization algorithm [1]. Also, the results were obtained using a partition size of (60%, 20%, 20%) for the training, validation and testing sets and the learning rate was selected using the validation set. Models were trained using Mixed precision, a NVIDIA extension enabling distributed training for Pytorch. Tables containing information about the architectures in a more condensed manner can be found in App. B.

## 5 Reproduction Results

In this section, we present our reproduction results for the Supervised and Unsupervised experiments. We followed as closely the ideas presented by the authors but as previously mentioned the default dataset of size 1 million nor the standard deviation on some results (where we only reproduced one seed) were reported. Because the standard deviation was relatively small and the default dataset followed the same data distribution, we believe our overall conclusion on the reproducibility still holds.

### 5.1 Supervised Learning

The results of supervised learning experiment were used as the baseline for all the other experiments presented in the article. For this reason it seemed imperative for us to start by reproducing those results. Here are the results we obtained, see table 1.

| Dataset Size | Synbols Default 100k | Camouflage 100k | Korean 100k | Less Variation 100k |
|---|---|---|---|---|
| MLP | 14.56 +0.27 | 3.98 +0.10 | 0.11 +0.1 | 0.06 +0.05 |
| Conv-4-Flat | 68.47 +0.04 | 34.62 -2.27 | 2.07 -0.45 | 0.22 -0.01 |
| Conv-4-GAP | 70.83 -0.69 | 28.90 +0.70 | 33.96 -0.38 | 3.53 -0.37 |
| ResNet-12 | 95.58 -0.15 | 90.44 -0.30 | 96.92 +0.16 | 38.51 +0.9 |
| ResNet-12+ | 97.24 -0.08 | 94.39 -0.04 | 98.58 -0.04 | 57.63 -0.21 |
| WRN-28-4 | 93.74 -0.17 | 86.64 +0.30 | 96.47 -0.68 | 22.18 +0.92 |
| WRN-28-4+ | 97.38 -0.03 | 95.54 +0.01 | 99.27 -0.13 | 67.02 +1.4 |

Table 1: **Reproduction of Supervised Learning Results:** Accuracy of various models on supervised classification tasks. Deviation from original results are in gray.

We can see that the results are very similar to the results reported in the paper [3] confirming the assessment of the authors on the versatility of the synthetic data generated. While all models were able to achieve +98% accuracy on MNIST dataset, only the state-of-the-art models were able to achieve high accuracy on more sophisticated datasets generated by the tool. We can also assert that Synbols can be used to provide meaningful data augmentation [3], increasing by a factor of three the accuracy achieved on the hardest dataset (i.e Less Variations). In addition, we trained a second time the MLP using the same datasets generated on our own and obtained very similar results.

---

[1]Due to limited resources we were not able to run the larger Default variant dataset.

[2]Originally generated using three different pseudo random seed to replicate the results

[3]Here, data augmentation consists of uniformly sampled affine deformations in the attributes.

 **5.2 Unsupervised Representation Learning**

 The reproduction results are reported in the following table.

| | Character Accuracy | | | Font Accuracy | | |
|---|---|---|---|---|---|---|
| | **Solid Pattern** | **Shades** | **Camouflage** | **Solid Pattern** | **Shades** | **Camouflage** |
| **Deep InfoMax** | 82.69 +1.18 | 6.15 +0.37 | 5.48 -0.63 | 15.37 +1.07 | 0.23 +0.08 | 0.25 +0.03 |
| **VAE** | 60.73 +2.75 | 22.17 +0.26 | 2.98 +0.87 | 2.11 +0.57 | 0.27 +0.09 | 0.11 +0.07 |
| **HVAE** | 68.92 -2.2 | 28.32 -0.54 | 3.79 +0.12 | 1.9 +0.81 | 0.29 +0.1 | 0.16 +0.01 |

Table 2: **Reproduction of Unsupervised Representation Learning Results:** Accuracy of a MLP classifier downstream. Deviation from original results are in gray.

Again, we observe the reproduced results are aligned with the ones reported in the paper. Although all models perform well in character classification on the solid pattern dataset, we observe the same significant drop on the Shades and Camouflage variants. Those results are very different from the ones reported in the Supervised experiment. In Sec. 5.2 we mention some of our hypothesis regarding this issue.

# 6 Discussion of findings

## 6.1 Supervised Learning

The table shown in In Sec. 5.1 report the test set loss from our reproduction. However, it is still interesting to mention how fast different supervised learning models reduce the validation loss to the optimum through iterations of epoch. This can reflect the ability of models tackling the synthetic datasets. Specifically, except on the Less Variation dataset we noted that WRN performed really well on the classification task. We believe this model, thanks to its wider convolutional layers, benefits from the rich composition of latent features generated by Synbols. What is also impressive is the speed at which it achieves high accuracy and robustness (i.e Generalization). Even on the hardest dataset tested, the optimal training and validation losses were reached at the $25^{th}$ epoch as shown in Figure 1. [4]

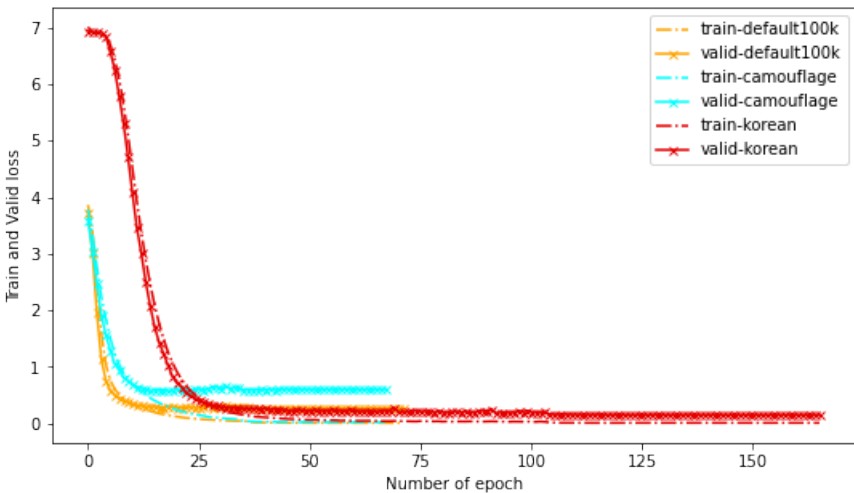

Figure 1: Cross Entropy loss of WRN on various datasets

We have noticed a difference in the choice of channels between what was reported in the paper and their code for Conv-4. From inspecting their code, we found that instead of the 64 channels for all layers claimed in the article, the 4 layers had 32, 64, 128, 256 channels respectively.

---

[4]The final test loss for each model is reported in Tab. 2.

## 6.2 Unsupervised Representation Learning

Although all the models were able to learn meaningful representations on the Solid dataset, a major drop was observed when adding Camouflage. The best performing model, Deep InfoMax for Solid and Camouflage Pattern was the least performing on Shades. It seems that due to the global structure of the gradient pattern Deep InfoMax, the model struggles to capture meaningful latent features in the limited size representation. Intuitively speaking, we believe that the local feature in the gradient pattern can be very different from the global feature of the original image [5] and this is why Deep InfoMax did not capture meaningful representations for Shades dataset.

We tried to increase the accuracy by performing a grid search on the MLP classifier downstream and also tried with a linear regression model, both methods lead to similar performance (5% margin).

## 7 Conclusion

Despite a couple of points that were different in the code from what was reported in the paper, we applaud authors reproducibility efforts and their availability when we had questions. We were able to reproduce the original results without major drawbacks. We thus conclude by answering the three key questions as followed;

- Are the original results reproducible? *Yes.*

- Were there any hidden assumptions in order to obtain the same results? *No.*

- Can we quickly generate similar datasets? *Yes.*

Synbols is a very versatile tool for rapidly generating rich composition of latent features in low resolution images effectively probing a wide range of machine learning algorithms. We also observe that it can help identify latent properties and increase the robustness of a model on smaller datasets.

Although its limited generation capabilities (i.e : UTF-8 symbols only), authors are planning to add more features to the current generator and also extend the concept to video generation/visual question answering support. We are very excited to see its impact on the computer vision field and hopefully on the whole field of deep learning.

## 8 Discussion

This report focuses on the reproduction of some results presented of the above-mentioned paper [3]. Authors introduced a new data generator called Synbols allowing fast generation of low-resolution images rich in latent features. Researchers explored the capabilities of the tool by training popular machine learning algorithms in various M.L paradigms with their synthetically generated datasets. The tool is also trying to address some broader issues relevant to the whole field (i.e.faster iteration cycles for the training, less reliance on expensive hardware, etc.). In this report, we follow the replication instructions and the published code provided by the authors in order to verify some of those claims. The final code used for implementing the replicated results can be found here: [Re] Synbols Repository.

### 8.1 What was easy

We applaud authors reproducibility efforts and their availability whenever we had questions. A repository specifically made in order to facilitate the reproduction was available and an up-to-date docker image was also at our disposal to help generate more datasets with the tool. No hidden/forgotten assumptions were needed to reproduce their results. Thanks to those all those efforts our task was significantly simplified.

### 8.2 What was difficult

Originally, for the two paradigms tested, twelve different models were trained. Although the important hyper-parameters were always mentioned or referenced, we sometimes struggled to check their implementation to see if everything was performed as reported. But authors always made time to explain implementation details that were more difficult to understand at first glance.

---

[5]The model is more likely to confuse gradient changes with important symbol information.

## 8.3  Communication with original authors

We actively reached out the original authors through e-mail and meeting sessions. The authors always made time to answer our questions. Hereby, we sincerely thank the authors for providing us adequate supports during the reproducibility.

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

# Appendix

## A. Supervised Learning

| MLP Parameters | Value |
| --- | --- |
| Layers | 3 |
| Hidden size | 256 |
| Activation | Leaky ReLU non-linearities |
| Learned parameters | 72k (fully connected) |

| Conv-4-GAP Parameters | Value |
| --- | --- |
| Convolution layers | 4 |
| Channels per layer | 64 |
| Pooling | Global average |
| Learned parameters | 112k |

| Resnet-12 Parameters | Value |
| --- | --- |
| Residual Layers | 12 |
| Residual blocks | 4 |
| Channel/Output per block | {64,128,256,512} |
| CNN per block | 3 |
| CNN structure | 3x3 |
| Activation | ReLU non-linearities |
| Pooling (at the end of each block) | Max |
| Dropout(first& second convolution at each block) | 0.1 |
| Learned parameters | 8M |

| WRN-28-4 Parameters | Value |
| --- | --- |
| Residual Layers | 28 |
| Residual blocks | {16,4,4,4} |
| Output per block | {16,32,64,128}*4 |
| CNN structure | 3x3 |
| Activation | ReLU non-linearities |
| Pooling | Global average |
| Dropout | 0.1 |
| Batch size | 128 |
| Learned parameters | 5.8M |

## B. Unsupervised Supervised Learning

| Deep InfoMax hyperparameters | Value |
|---|---|
| Seed | 2 |
| Dropout | 0.3 |
| Activation Function | ReLu |
| Kernel | 3 |
| Stride | 1 |
| Padding | 1 |
| Feature Vector Size | 64 |
| Global Discriminator Number of Convolutional Layers | 2 |
| alpha | 0.5 |
| Local Discriminator Number of Convolutional Layers | 3 |
| Beta | 1.0 |
| Prior Discriminator Number of Fully-Connected Layers | 3 |
| Gamma | 0.1 |

| Variational Auto-Encoder hyperparameters | Value |
|---|---|
| Dropout | 0.3 |
| Activation Function | leaky ReLu |
| Kernel | 3 |
| Stride | 1 |
| Padding | 1 |
| Pooling | 2x2 |
| Beta | 0.01 |
| Feature Vector Size | 64 |
| Hierarchichal = True for HVAE | False |

