# OpenReview forum: "[Re] Synbols : Probing Learning Algorithms with Synthetic Datasets"
_ML_Reproducibility_Challenge/2020 — Reject_

### Official Review · AnonReviewer2 · 2021-02-10
**A successful replication**

**Rating:** 7
**Confidence:** 4

**Review:**

The manuscript presents a replication study on the Lacoste, A. et. al paper “Synbols: Probing Learning Algorithms with Synthetic Datasets” which presents a tool for generating images of unicode symbols with control over parameters like font, language, resolution, background texture, etc. Paper also presents experiments showing use of Synbols in standard supervised learning setting (establishing baselines), in out-of-distribution testing, for active learning, unsupervised representation learning, and object counting. The authors of this report have been able to back up the results from the original paper.

- The report clearly define and describe the experimental setting of the original paper. The overall organisation and clarity of the document is excellent.
- The authors were able to successfully implement both the proposed algorithms from the description of the algorithms in the original paper/appendix (they had a fluid communication with the original authors for testing reproducibility). Also, the implementation of all the reproduced experiments is provided in a github repository, which is remarkable.
 - The report contains a summarised discussion on the state of reproducibility of the original paper. Results obtained are very close to the values reported by the original authors.


Weak points:

- The authors were unable to (completely) run (and check) the original experimental setting for some DNN models due computational constraints (limiting their analysis to just one shot/seed). Still, results obtained in those cases are still similar to those in the original paper.
- Not running all the datasets (e.g., 1M) when analysing the amortised times makes the analysis a bit incomplete. Perhaps Authors should have had a look for other compute resources (codeOcean, Kaggle, etc.).

Minor:

- Typo in footnote 2.
- Typo in Methodology: "using only seed of the same dataset"


**Familiar With The Original Paper:**

I have read the original paper

**Reproducibility Summary:**

Report has summary

---

### Official Review · AnonReviewer1 · 2021-03-03

**Rating:** 6
**Confidence:** 4

**Review:**

The authors of this report reproduce part of the original paper (OP), and add a few experiments which test the quality of learned representations by trying different downstream classifiers. The authors found minor discrepancies between the hyperparameters reported in the OP and those contained in the code.

Format: the authors did not follow the provided latex template.
Scope: the authors reproduce some of the results from the OP
Code: at a glance the provided code appears complete. The authors mostly reuse the code provided in the OP, with some additions. It seems that the added code contains implementations copied from other open-source repositories. This is fine, but it is polite to explicitly reference those in the Readme.
Communication & hyperparameters: the authors have done their due diligence
Ablation/extensions: the authors add a few experiments which are consistent with the results of the OP. It would have been very surprising for these experiments to deviate from the existing results, and the motivation for this choice is unclear. It would have been more interesting to find, e.g. settings where the dataset generation fails, is computationally hard, or makes no sense.
Discussion: the authors make appropriate discussions and remarks on reproducibility, basically confirming that the material provided with the OP makes things easily reproducible.

- it is unusual to report the validation loss, which is used to tune hyperparameters. The test loss is normally reported instead.
- it would be good to clearly separate what was originally done in the OP from what was reproduced, and from what was added (not in the OP).
- the text is well structured and readable.


**Familiar With The Original Paper:**

I have not read the original paper

**Reproducibility Summary:**

Report has summary

---

### Decision · Program_Chairs · 2021-03-31

**Decision:**

Reject

**Comment:**

Overall reviews and/or the paper content not good enough for the AC to recommend to the journal.